# Mesoporous Silica Particle as an RNA Adsorbent for Facile Purification of In Vitro-Transcribed RNA

**DOI:** 10.3390/ijms241512408

**Published:** 2023-08-03

**Authors:** Eunbin Cho, Jayoung Namgung, Jong Sam Lee, Jinmin Jang, Bong-Hyun Jun, Dong-Eun Kim

**Affiliations:** Department of Bioscience and Biotechnology, Konkuk University, 120 Neundong-ro, Gwangjin-gu, Seoul 05029, Republic of Korea; 7bini@naver.com (E.C.); nagu9@hanmail.net (J.N.); znznfn2378@naver.com (J.S.L.); jjm6665@naver.com (J.J.); bjun@konkuk.ac.kr (B.-H.J.)

**Keywords:** RNA purification, in vitro transcription, mesoporous silica, spermidine

## Abstract

Messenger RNA vaccines against SARS-CoV-2 hold great promise for the treatment of a wide range of diseases by using mRNA as a tool for generating vaccination antigens as well as therapeutic proteins in vivo. Increasing interest in mRNA preparation warrants reliable methods for in vitro transcription (IVT) of mRNA, which must entail the elimination of surplus side products such as immunogenic double-stranded RNA (dsRNA). We developed a facile method for the removal of dsRNA from in vitro transcribed RNA with mesoporous silica particles as RNA adsorbents. Various polyamines were tested for the facilitation of RNA adsorption onto mesoporous silica particles in the chromatography. Among the polyamines tested for RNA adsorption, spermidine showed a superior capability of RNA binding to the silica matrix. Mesoporous silica-adsorbed RNA was readily desorbed with elution buffer containing either salt, EDTA, or urea, possibly by disrupting electrostatic interaction and hydrogen bonding between RNA and the silica matrix. Purification of IVT RNA was enabled with the adsorption of RNA to mesoporous silica in a spermidine-containing buffer and subsequent elution with EDTA. By differing EDTA concentration in the eluting buffer, we demonstrated that at least 80% of the dsRNA can be removed from the mesoporous silica-adsorbed RNA. When compared with the cellulose-based removal of dsRNA from IVT RNA, the mesoporous silica-based purification of IVT RNA using spermidine and EDTA in binding and elution, respectively, exhibited more effective removal of dsRNA contaminants from IVT RNA. Thus, mRNA purification with mesoporous silica particles as RNA adsorbents is applicable for the facile preparation of nonimmunogenic RNA suitable for in vivo uses.

## 1. Introduction

The success of mRNA-based vaccines, particularly during the COVID-19 pandemic, has demonstrated their potential to address other infectious diseases and cancers [1]. mRNA therapy offers advantages such as rapid production, scalability, and customization of specific pathogens or variants [2]. The versatility and potential for customization of mRNA therapies have opened new possibilities for addressing a wide range of diseases [3]. Consequently, the demand for effective techniques for purifying nonimmunogenic mRNA from in vitro transcription (IVT) has steadily increased [4,5,6]. mRNA production is initiated by designing a DNA template containing the T7 promoter sequence, which is then transcribed into mRNA using T7 RNA polymerase [7]. The inherent inaccuracy of T7 RNA polymerase during enzymatic transcription results in the production of double-stranded RNA (dsRNA) impurities. These contaminants may arise due to several reasons, including random priming of abortive transcripts [8], the occurrence of antisense transcription, and turn-around transcription [9,10]. The presence of dsRNA contaminants in synthesized mRNA can have undesirable effects in downstream applications, such as interference with protein expression and triggering innate immune responses [11,12]. To enhance the translational quality of mRNA and mitigate potential detrimental effects, it is crucial to remove dsRNA contaminants from in vitro-transcribed mRNA [13].

There are two main approaches for removing dsRNA: minimizing the generation of dsRNA during the IVT process and eliminating dsRNA after the IVT process. To decrease dsRNA synthesis, previous studies have utilized various techniques such as competing oligonucleotides [14,15], chaotropic agents [16], or high temperatures [17]. However, the use of custom reagents or elevated temperatures poses substantial economic and technical challenges when scaling up mRNA manufacturing processes [18]. Another approach is to use engineered T7 RNA polymerase, which can reduce the formation of dsRNA byproducts [18,19,20]. Nevertheless, the complexity of the engineering procedure and the limited commercialization of the modified T7 RNA polymerase pose a challenge in terms of cost-effectiveness for its implementation in many laboratories. Posttranscriptional purification methods include chromatographic techniques such as oligo(dT)-coupled, bead-based affinity chromatography, reverse-phase, high-performance liquid chromatography (HPLC), and cellulose-based isolation for the separation of dsRNA [21,22]. Despite its extensive uses in the mRNA production process for removing dsRNA contaminants, ion-pair reverse phase HPLC using triethylammonium acetate (TEAA) and acetonitrile has limitations in terms of scalability and use of toxic solvents, such as acetonitrile [16,17,18,19,20].

Silica is widely used for nucleic acid purification because its negatively charged surface contains silanol groups [23]. Among the different types of silica, mesoporous silica provides a large surface area for efficient adsorption due to its well-defined and ordered pore structure [24]. These characteristics enable effective binding and separation of nucleic acids, making mesoporous silica a valuable tool for nucleic acid purification [25,26]. There are two methods of adsorbing nucleic acids onto silica surfaces. The first method uses high salt concentrations and chaotropic agents [27]. However, the use of chaotropic agents has limitations, including their potential for nucleic acid degradation and the need for larger volumes of chaotropic salts for scalability. The second method involves functionalizing the silica surface with polyamine molecules, which enables electrostatic interactions with nucleic acids [28]. Although this method avoids the use of chaotropic agents, the synthesis of polyamine-functionalized silica is complex and may require multiple steps, affecting the overall cost and scalability of the process.

Herein, we developed a facile chromatography method for RNA purification using mesoporous silica as an adsorbent and polyamines in the mobile phase, avoiding the use of hazardous chaotropic salts or functionalized silica for RNA purification. We observed that spermidine, a positively charged polyamine present in the IVT reaction buffer, could lead to the adsorption of RNA onto mesoporous silica. We explored not only the capability of spermidine-based RNA purification but also the removal of dsRNA content in the purified IVT RNA. The spermidine-based RNA purification method provides a straightforward and nonhazardous approach for isolating nonimmunogenic RNA.

## 2. Results and Discussion

### 2.1. Adsorption of IVT RNA to Silica in the Presence of Spermidine

To investigate whether IVT RNA binds to mesoporous silica, we first mixed IVT RNA with mesoporous silica dispersed in a buffer containing sodium, magnesium, or calcium salts in a microcentrifuge tube (Figure 1A). These cations are considered potential facilitators of nucleic acid binding because of their direct involvement in the formation of nucleic acid−silica complexes [29]. After mixing IVT RNA with mesoporous silica particles in a batch mode for 10 min, we added these ions to the 2μg RNA and mesoporous silica. Following incubation for 10 min, centrifugation was performed to separate the unbound IVT RNA from mesoporous silica. The supernatant was collected and analyzed for the presence of RNAs by denaturing urea-PAGE (6%). When the three cations were added, most of the RNAs were detected in the supernatant, indicating that these ions did not facilitate the binding of IVT RNA to the mesoporous silica surface (Figure 1B).

Unexpectedly, when IVT RNAs were mixed with the IVT buffer, we observed that IVT RNA remained in the pellet after the centrifugation. This intriguing result led us to hypothesize that the substance(s) present in the transcription IVT buffer may facilitate the binding of RNA to mesoporous silica. To find out which substance in the IVT buffer induces the binding of RNA to mesoporous silica, various combinations of MgCl2, dithiothreitol, and spermidine in the IVT buffer were tested for facilitation of IVT RNA binding to the mesoporous silica (Figure 1C). When spermidine was added to the IVT RNA solution, IVT RNA was found to remain in the pellet due to the binding of RNA to the silica particles. This result suggests that spermidine functions as a binding agent to facilitate the adsorption of IVT RNA onto mesoporous silica.

Next, we investigated the amount of spermidine required to induce the binding of RNA to the mesoporous silica. To determine the amount of spermidine necessary to bind IVT RNA adsorption onto mesoporous silica, we examined the N/P ratio between the cationic amine groups (N) and negatively charged phosphate groups (P) in RNA. Starting with an N/P ratio of 20, we systematically reduced the N/P ratio by half to determine the minimum N/P ratio needed for IVT RNA to bind to the silica surface (Figure 1D). IVT RNAs were significantly adsorbed onto the mesoporous silica when the spermidine was provided in the buffer at the N/P ratio of above five, while IVT RNA failed to be bound to the silica at a reduced N/P ratio below five.

### 2.2. Characteristics of Polyamine for RNA Purification with Mesoporous Silica-Based Chromatography

Having established that spermidine can facilitate the adsorption of IVT RNA to silica particles in a batch mode of chromatography, we tested the feasibility of applying silica particles to fractionate IVT RNA in a simple separation chromatography using spin columns filled with mesoporous silica particles (Figure 2A). We first compared the binding of RNA to mesoporous silica when spermidine was either present in the buffer, as for the mobile phase, or immobilized by conjugation onto silica in the stationary phase (Figure 2B). From the flowthrough fraction, most of the input RNA has been eluted as unbound RNA under various concentrations of spermidine present in the stationary phase. In contrast, most of the input RNA was bound to the silica particles without unbound RNA in the flowthrough when the spermidine at the N/P ratio above five was included in the binding buffer as mobile. This result indicates that spermidine has a more pronounced effect on RNA binding when it is in the mobile phase than when immobilized in the stationary phase.

Nucleic acid adsorption to silica is likely attributed to diverse driving forces, such as electrostatic interaction, free electrolytes in solution, and hydrogen bonding between single-stranded RNA and the silica surface [26]. To explore the properties of IVT RNA adsorption onto mesoporous silica in the presence of spermidine, several elution buffers, including salt (NaCl), ethylenediaminetetraacetic acid (EDTA), or urea, were applied for the capability of dissociating the RNA bound to mesoporous silica (Figure 2C). High salt concentration and EDTA in the elution buffer effectively eluted the IVT RNA bound to the mesoporous silica in the presence of spermidine in the binding buffer. This result suggests that silica is capable of negatively charging RNA through cationic polyamine (i.e., spermidine) as an electrostatic linkage between RNA and the mesoporous silica surface. EDTA is likely stripping out RNA-bound spermidine by chelating the positively charged spermidine with carboxylate groups in EDTA. Interestingly, RNA elution was observed with a high concentration of urea, a chaotropic agent that disrupts hydrogen bonding, indicating that spermidine-linkage between RNA and mesoporous silica is also partially driven by hydrogen bonding.

Next, we examined the effects of three linear polyamines (ethylenediamine, spermidine, and spermine) on RNA binding to mesoporous silica (Figure 2D). Most input RNAs were eluted in the flowthrough fraction as unbound RNA in the presence of ethylenediamine, whereas spermidine and spermine in the binding buffer allowed the input RNAs to be retained as bound to the mesoporous silica. Thus, spermidine and spermine are more effective than ethylenediamine in binding RNA to silica, which is likely caused by the involvement of the secondary amine group(s) in the polyamine for linking RNA to the silica surface. Furthermore, the desorption of RNA bound to mesoporous silica in the presence of spermidine or spermine was tested by eluting the adsorbed RNA with an elution buffer containing NaCl (300 mM). The silica-bound RNAs in the presence of spermidine less than 4 mM (equivalent to the N/P ratio of 40) were readily eluted with the elution buffer (Figure 2D). In contrast, the silica-bound RNAs with spermine were marginally eluted with the elution buffer at a low concentration of spermine of 1 mM (equivalent to the N/P ratio of 13). This result indicates that the association of RNA with the silica in the presence of linear polyamine is likely affected by the number and position of amines with the N/P ratio; four amine groups in spermine probably form robust electrostatic interactions between RNA and silica, resulting in hindered disruption of RNA-mesoporous, silica-spermine complexes during salt elution. Thus, the binding of IVT RNA to mesoporous silica in the spermidine-containing buffer and facile desorption of the RNA with electrolyte are unique properties of linear polyamines harboring secondary amine to bridge negatively charged RNA to silica material.

### 2.3. Mesoporous Silica Rather than Spherical Silica with EDTA Elution as Facile Adsorption/Desorption of RNA

To examine the effect of silica structure on RNA adsorption and desorption with electrolyte elution buffer, binding and elution efficiencies of RNA with mesoporous silica and spherical silica were compared (Figure 3A). We employed mesoporous silica of the SBA-15 (Santa Barbara Amorphous-15) type, which is characterized by particles with a size of less than 150 μm and well-defined pores with a diameter of 12 nm. The highly ordered hexagonal pattern of these pores results in a surface area of up to 1000 m^2^/g [30]. The amount of spermidine in the RNA binding buffer was gradually increased to compare the RNA binding capability of the mesoporous and spherical silica. We observed that a significant amount of RNA was present in the flowthrough as unbound RNA to silica when RNA was incubated with spherical silica in the presence of spermidine at varying concentrations. In contrast, when mesoporous silica was used in the chromatography, the amount of RNA in the flowthrough decreased as the amount of spermidine increased, indicating that the input RNA was effectively bound to mesoporous silica in the presence of spermidine. The amount of recovered RNA from spherical silica with electrolyte elution was significantly lower than that incubated with the same amount of spermidine added to the mesoporous silica (Figure 3B). These results demonstrate that mesoporous silica is more effective than spherical silica in the adsorption and desorption of RNA in silica-based chromatography.

Next, to enhance the recovery rate of RNA bound to the silica, we compared salt (i.e., NaCl) and chelating reagent (i.e., EDTA) for elution of the silica-bound RNA during the RNA purification process (Figure 4A). Stepwise elution with gradually increasing concentrations of EDTA or NaCl was applied to the RNA-adsorbed mesoporous silica column. As the concentration of EDTA or NaCl increased, the amount of RNA eluted from the mesoporous silica increased. The analysis of eluant for the presence of eluted RNAs with agarose gel electrophoresis showed that more than 80% of the input RNA was recovered with EDTA elution (Figure 4B). However, when the silica-bound RNA was eluted with a gradual increase in salt concentration, the combined amount of recovered RNA in the eluants did not reach 20% of the input RNA. Thus, EDTA elution has a superior capacity for RNA recovery to electrolyte elution when used in a spermidine-mediated, RNA-mesoporous silica binding system.

After confirming the effectiveness of EDTA as an elution solution, we examined the amount of EDTA required for the elution of a given amount of RNA adsorbed to the mesoporous silica in the presence of spermidine (Figure 4C). Various amounts of RNA were loaded onto the mesoporous silica spin column in the presence of spermidine at the N/P ratio of 20, which was then eluted with a stepwise increase of EDTA amount (500 and 1000 nmol). The analysis of the eluted fractions by agarose gel electrophoresis demonstrated that loaded RNA with an amount less than 10 μg was effectively eluted by EDTA elution at 500 nmol, while most of the loaded RNA with 20 μg amount was not eluted with 500 nmol of EDTA but eluted with a higher amount of EDTA (1000 nmol). These findings indicate that the amount of EDTA required for the elution of a given amount of RNA in the spermidine-mediated, RNA-mesoporous silica binding system needs to be adjusted in proportion to the amount of RNA loaded in the chromatography.

Next, we tested whether the spermidine-mediated, RNA-mesoporous silica system is compatible with unprecipitated IVT RNA by directly loading a post-IVT reaction mixture onto the mesoporous silica chromatography. Purification of RNA directly from a post-IVT reaction solution could be challenging because various substances such as enzymes and free nucleotides in the IVT reaction mixture can interfere with the binding of RNA to the matrix and subsequent elution steps, thus impacting purification efficiency and RNA integrity [6]. The analysis of flowthrough fraction after loading the post-IVT reaction onto the mesoporous silica column by agarose gel electrophoresis demonstrated that synthesized RNA in the IVT reaction containing spermidine readily bound to the mesoporous silica in the presence of various reaction reagents in the mixture of IVT reaction solution (Appendix A). In addition, to assess the size, purity, and quantitation of the input IVT RNA and the EDTA-eluted RNA after the mesoporous silica-filled spin column chromatography, we analyzed the integrity of RNA using the capillary electrophoresis system with a high level of accuracy and precision (Figure 4D). The analysis of the RNA eluted from the mesoporous silica chromatography by an electropherogram and elution profile confirmed that the EDTA-eluted IVT RNA through the spermidine-mediated mesoporous silica spin column chromatography remains intact in quality (i.e., a single RNA band with the same size) as well as in quantity with good recovery (i.e., the same height of peaks in the elution profile).

### 2.4. Removal of dsRNA from IVT RNA Using Mesoporous Silica-Based Chromatography

To test whether the spermidine-mediated mesoporous silica spin column chromatography is applicable for the removal of dsRNA contaminants from IVT RNA, we performed the mesoporous silica-filled spin column chromatography with stepwise elution by increasing the EDTA concentration (Figure 5). We evaluated our mesoporous silica RNA purification system by comparing dsRNA removal efficiency with the previously established dsRNA removal system using cellulose chromatography [22]. Based on our observation that 10 μg of the RNA sample required 500 nmol of EDTA for elution in the mesoporous silica-filled spin column chromatography (Figure 4C), we first added 100 μL of 5 mM EDTA to the RNA-loaded spin column and gradually increased the concentration of EDTA. The analysis of the eluant with the quantification of dsRNA using ELISA indicates that the RNA eluted with 5 and 6 mM EDTA had a lower proportion of dsRNA content than the initial input RNA (Figure 5A). Especially, the proportion of dsRNA content in the eluant obtained by elution with 5 mM EDTA was comparable to that of the eluant obtained using cellulose chromatography with 16% (*v*/*v*) ethanol.

To further confirm that the RNA eluted with 5 mM EDTA had a lower dsRNA content than the input RNA, the dot blotting analysis was performed with J2 dsRNA-specific antibody (Figure 5B). Equal amounts of RNA from each eluant were analyzed by agarose gel electrophoresis and the dot blotting assay for quantification of input RNA and dsRNA, respectively. The quantification of the dsRNA proportion indicates that the RNA fraction eluted with 5 mM EDTA from the mesoporous silica-filled spin column chromatography contained approximately 90% less dsRNA than the corresponding input RNA. In contrast, the elution with 8 mM EDTA was not as effective as 5 mM EDTA elution for dsRNA removal. Importantly, J2 dsRNA-specific antibody analysis of the RNA fraction eluted with 5 mM EDTA and the RNA fraction eluted with 16% (*v*/*v*) ethanol from the cellulose demonstrated that the efficacy of dsRNA removal by both purification methods was almost comparable (94% vs. 92%, respectively).

Next, we analyzed the elution profiles of total RNA and dsRNA in each eluant by using the RiboGreen assay and dsRNA ELISA, respectively (Figure 5C). The analysis of the relative proportion of dsRNA in each eluted RNA fraction with increasing EDTA concentrations revealed that dsRNA exhibited a broader elution pattern than total RNA eluted in gradual elution with increasing EDTA concentration, suggesting that dsRNA bound to mesoporous silica more tightly than ssRNA in the presence of spermidine. One possible explanation for this observation is that the two complementary phosphate backbones in dsRNA allow for more electrostatic interactions with both the mesoporous silica surface and amine groups of spermidines than with ssRNA. Fewer phosphate backbones in single-stranded RNA may lead to fewer potential interactions with spermidine and the mesoporous silica surface, resulting in a weaker binding affinity than that of dsRNA. This conjecture is consistent with our observation that urea as a chaotropic reagent by denaturing dsRNA into ssRNA can elute the silica-bound RNA (Figure 2C), in which ssRNA has a weaker affinity to the silica as compared to the corresponding dsRNA. Notably, the RNA recovery rate when using the mesoporous silica to elute RNA with 5 mM EDTA was approximately 40% of the input RNA, whereas the recovery rate was ~30% with the cellulose resin with 16% (*v*/*v*) ethanol elution (Figure 5C). This result indicates that RNA eluted with 5 mM EDTA had a higher yield of RNA than that eluted with ethanol in the cellulose-filled spin column chromatography. Using the recovery rate of total RNA and the proportion of dsRNA in the fraction eluted with 5 mM EDTA, the percentage of dsRNA removed from the initial input dsRNA in the total RNA was obtained to be at least 80%. Taken together, we suggest that the elution of RNA with EDTA using the spermidine-based mesoporous silica chromatography is as efficient as cellulose-based chromatography for the purification of IVT RNA with the removal of dsRNA contaminants.

## 3. Materials and Methods

### 3.1. In Vitro Transcription of RNA

The plasmid DNA encoding EGFP (obtained from Addgene, Watertown, MA, USA) harboring the 3’ UTR, 5’ UTR, and poly(A) tail sequences was linearized by digestion with EcoRI restriction enzyme (Takara, Tokyo, Japan). Some 2 μg of the linearized plasmid was mixed in 50 μL reaction solution containing 2 mM rants mixture (Promega, Madison, WI, USA), 100 U T7 RNA polymerase (Thermo Fisher Scientific, Waltham, MA, USA), and 40 U recombinant RNase inhibitor (Takara) in 1 × T7 RNA polymerase buffer containing 50 mM Tris-HCl (pH 7.5), 15 mM MgCl2, 2 mM spermidine (Sigma-Aldrich, St. Louis, MO, USA), and 5 mM DTT (Sigma-Aldrich). The in vitro transcription reaction mixture was incubated at 37 °C for 2 h, and 5 U DNase I (Takara) was added for DNA degradation after 2 h incubation. After incubation at 37 °C for 30 min, 20 mM EDTA was added to the reaction mixture to stop enzymatic reactions. Following the addition of EDTA, the synthesized RNA was isolated by precipitation with 8 M LiCl solution (Sigma-Aldrich) and subsequently dissolved in nuclease-free water. The concentration of the LiCl purified IVT RNA was measured using a UV–Vis spectrophotometer (Ultrospec 2100 pro spectrophotometer; Biochrom Ltd., Cambridge, UK) at a wavelength of 260 nm. Aliquots of the LiCl purified IVT RNAs were heat-denatured and then separated by electrophoresis on 1% agarose gels containing 0.01% (*v*/*v*) GelRed nucleic acid gel stain (Biotium, San Francisco, CA, USA). RNA bands in the agarose gel were visualized using a UV transilluminator. Alternatively, for analysis of RNA using the capillary electrophoresis system, RNA was analyzed using an Agilent 4150 TapeStation (Agilent Technologies, Santa Clara, CA, USA) with RNA ScreenTape (Agilent Technologies) according to the manufacturer’s instructions.

### 3.2. Calculation of the N/P Ratio

The N/P ratio, representing the number of amine groups (N) in spermidine relative to the number of bases in the phosphate backbone (P) of RNA was calculated by determining the moles of RNA and spermidine. The mass of the in vitro transcribed RNA (996 nts) was divided by the molecular weight of RNA (320 kDa) to calculate the number of moles of RNA in the sample. Multiplying the number of moles of RNA by the number of nucleotides in the RNA backbone (996 nts) gives the total number of phosphate backbones (P) present in the RNA sample. The number of amine groups (N) in spermidine was obtained by multiplying the moles of spermidine by three (three amine groups per mole of spermidine). The N/P ratio was then obtained by dividing the number of amine groups (N) by the number of phosphates in the RNA (P).

### 3.3. Purification of IVT RNA Using Mesoporous Silica-Based Chromatography

Mesoporous silica (SBA-15, cat. #913855, Sigma-Aldrich) slurry was prepared in 10 mM HEPES (pH 7.2) at a concentration of 30 mg/mL. Some 0.6 mg of mesoporous silica, the LiCl purified IVT RNA (2 μg) with 20 mM of various salts (NaCl, MgCl2, and CaCl2), or 1 × T7 RNA polymerase buffer (50 mM Tris-HCl (pH 7.5), 15 mM MgCl2, 2 mM spermidine, and 5 mM DTT) were mixed in a volume of 20 μL in a 1.5 mL microcentrifuge tube. The tube was incubated for 10 min and centrifuged at 3000× *g* for 3 s to separate the supernatant from the pellet. The collected supernatant was analyzed using urea-PAGE (6%) and visualized using a UV transilluminator after staining with SYBR Gold (Invitrogen, Waltham, MA, USA).

To perform RNA binding and elution experiments using a spin column, 20 μL of the mesoporous silica slurry (30 mg/mL) or spherical silica (cat. #78991, Sigma-Aldrich) slurry (30 mg/mL) in 10 mM HEPES (pH 7.2) was transferred to a Pierce™ spin column (Thermo Fisher Scientific) and centrifuged at 3000× *g* for 3 s. The LiCl purified IVT RNA (2 μg) or unpurified IVT RNA in the post-IVT reaction solution were mixed with spermidine at the N/P ratio of 20, 10 mM HEPES buffer (pH 7.2), and 10 mM NaCl in a volume of 20 μL. The RNA samples were transferred to a spin column containing mesoporous silica and incubated at 25 °C with shaking (20 rpm) for 10 min. The spin column was then centrifuged at 3000× *g* for 3 s to separate the unbound RNA from the mesoporous silica. Elution buffer containing designated concentrations of NaCl (Bio Basic, Toronto, Canada), EDTA (Bio Basic), or urea (Bio Basic) in 10 mM HEPES (pH 7.2) was added to the spin column in a volume of 20 μL. The spin column was then shaken for 10 min and centrifuged at 3000× *g* for 3 s to collect the eluant. For the stepwise elution with increasing the concentration of elution solution, 10 μg of the unpurified IVT RNA in the post-IVT reaction solution was mixed with spermidine at the N/P ratio of 20 in 10 mM HEPES (pH 7.2) and 10 mM NaCl. Next, 100 μL of 5 mM EDTA solution was added to the spin column and incubated for 5 min. Stepwise elution was carried out by consecutively applying an elution solution of EDTA with gradually increased concentration (5 to 10 mM). All eluants obtained from the spin column were loaded onto a 1% agarose gel containing 0.01% (*v*/*v*) GelRed and further analyzed with a UV transilluminator.

### 3.4. Spermidine-Conjugated Mesoporous Silica for Purification of IVT RNA

Spermidine-conjugated mesoporous silica particles were prepared as follows: 50 mg of mesoporous silica particles (SBA-15, Sigma-Aldrich) were amine-functionalized with 80 μL of 3-aminopropyl triethoxysilane (APTS, Siga-Aldrich) and 50 μL of saturated ammonium hydroxide solution (Sigma-Aldrich) in 2 mL of ethanol. After overnight shaking at 25 °C, the silica particles were first washed with 2 mL of ethanol, followed by two washes with 2 mL of methanol, and then dried in vacuo. Next, the amine-functionalized mesoporous silica particles were dispersed with 1.2 mmol of succinic anhydride (Sigma-Aldrich) and 1.2 mmol of N, N-diisopropylethylamine (DIEA, Sigma-Aldrich) in 3 mL of dimethyl formamide (DMF, Samchun Chemical, Seoul, Korea). After succinylation for 3 h with shaking at 25 °C, the silica particles were washed with 2 mL of DMF, methylene chloride (MC, Daejung Chemicals, Gyeonggi-do, Korea), methanol, and dried in vacuo. Lastly, to conjugate spermidine with the silica particles, 0.4 mmol of diisopropylcarbodiimide (DIC, Sigma-Aldrich), 0.4 mmol of hydroxy benzotriazole (HOBT, Sigma-Aldrich), 0.48 mmol of DIEA, and 0.48 mmol of spermidine solved in 1.5 mL of DMF were added to succinylate mesoporous silica particles dispersed in 1.5 mL of DMF. The reaction proceeded with shaking at 25 °C overnight, and the silica particles were washed with 2 mL of DMF, MC, methanol, and dried in vacuo. The loaded amount of spermidine to mesoporous silica was determined to be 0.69 mmol/g by measuring the weight increment of silica particles.

The slurry of spermidine-conjugated mesoporous was prepared at a concentration of 30 mg/mL in 10 mM HEPES (pH 7.2). The LiCl purified IVT RNA (0.2 μg) was mixed with the spermidine-conjugated mesoporous silica slurry in a volume of 20 μL. The same amount of spermidine as that in the stationary phase was added to the mobile phase of the comparison group. RNA samples were incubated with shaking (rpm 20) at 25 °C for 10 min. The spin column was centrifuged at 3000× *g* for 3 s, and the eluants were loaded onto a 1% agarose gel containing 0.01% (*v*/*v*) GelRed. The relative amount of RNA was quantified using the ImageJ software (version 1.53e) based on the image of agarose gel under UV visualization.

### 3.5. Cellulose-Based Purification of IVT RNA

Cellulose fibers (cat. #C6288, Sigma-Aldrich) at a concentration of 0.2 g/mL were suspended in the chromatography buffer containing 10 mM HEPES (pH 7.2), 0.1 mM EDTA, 125 mM NaCl, and 16% (*v*/*v*) ethanol. After mixing for 10 min, 140 μL of the cellulose suspension was transferred to a Pierce™ spin column and centrifuged for 3 s at 3000× *g*. The flowthrough was discarded, and 100 μL of chromatography buffer was added to the cellulose fibers, followed by shaking for 10 min. Subsequently, the spin column was centrifuged (3000× *g*) for 3 s at 25 °C. The LiCl purified IVT RNA (10 μg) in 100 μL of chromatography buffer was added to the cellulose-filled spin column and agitated for 10 min to enable the binding of dsRNA to cellulose. Separation of unbound RNA from cellulose was achieved by centrifugation for 3 s at 3000× *g*.

### 3.6. RiboGreen Assay for RNA Quantification

Quantification of fractionated RNA was performed using a Quant-it™ RiboGreen RNA assay kit (Invitrogen). The RNA samples were diluted 25-fold in Tris-EDTA buffer (10 mM Tris-HCl (pH 7.5), 1 mM EDTA), and the RiboGreen dye was diluted 300 times in the Tris-EDTA buffer. The 25 μL of diluted RNA samples were mixed with an equal volume of diluted RiboGreen dye in a 96-well microplate and incubated for 5 min at 25 °C in the dark. Fluorescence measurements were performed using a VICTOR X3 Multilabel Plate Reader (PerkinElmer, Waltham, MA, USA) at excitation and emission wavelengths of 485 and 535 nm, respectively. The RNA concentrations in the samples were determined by comparing their fluorescence intensities with a standard curve, according to the manufacturer’s instructions. The assay was performed in triplicate for each RNA sample, and the average values were calculated.

### 3.7. Quantification of dsRNA by ELISA

dsRNA was quantified by sandwich ELISA using J2 dsRNA-specific, antibody-based dsRNA ELISA kit (Exalpha Biologicals, Shirley, MA, USA). J2 mouse monoclonal antibodies (included in the dsRNA ELISA kit) were immobilized on 96-well Nunc Immunoplates (Thermo Fisher Scientific) overnight at 4 °C and blocked with 1% (*w*/*v*) BSA in PBS for 2 h at 37 °C. The plate was then washed with PBS containing 0.5% (*v*/*v*) Tween 20 (Sigma-Aldrich). The 10 μL of RNA samples mixed with 90 μL of STE buffer (10 mM Tris-HCl (pH 7.5), 1 mM EDTA, 100 mM NaCl) were added to the plates and incubated for 1 h at 37 °C. After washing with PBS containing 0.5% (*v*/*v*) Tween 20 for four times, K2 mouse monoclonal antibody (included in the dsRNA ELISA kit) was added to the plate and incubated for 1 h at 37 °C. After washing four times, the plates were incubated with horseradish peroxidase-conjugated goat antimouse antibody (included in the dsRNA ELISA kit) for 1 h at 37 °C. After the washing, TMB substrate solution was added and incubated for 1 h at 25 °C in the dark. Some 2 M H_2_SO_4_ was added to stop the reaction, and the signal was read at 450 nm using a VICTOR X3 Multilabel Plate Reader. The amount of dsRNA in the test sample was determined from a standard curve generated using known amounts of dsRNA standard.

### 3.8. Dot Blotting Analysis for dsRNA Detection

For dot blotting assay using J2 dsRNA-specific antibodies, the nylon membrane (Cytiva Korea, Seoul, Korea) was placed on a sheet of 3M paper, and the silicone mask was tightly pressed onto the membrane to ensure that all wells were sealed to avoid sample leakage before loading the samples. A total of 100 ng of RNA samples were blotted onto nylon membranes using a 96-well biodot silicon gasket (Bio-Rad, Munich, Germany). After loading the samples, the membranes were air-dried and blocked in 5% (*w*/*v*) non-fat dry milk in TBS-T buffer (25 mM Tris-HCl (pH 7.5), 150 mM NaCl, 2 mM KCl, and 0.1% (*v*/*v*) Tween 20) at 25 °C for 1 h. The membrane was incubated with dsRNA-specific J2 monoclonal antibody (Sigma-Aldrich) diluted to a ratio of 1:5000. The antibody was diluted in 1% (*w*/*v*) non-fat dried milk and TBS-T buffer and incubated with the membranes on a rocker shaker at 4 °C overnight. The membranes were washed three times with TBS-T buffer for 10 min and then incubated with horseradish peroxidase (HRP)-conjugated antimouse immunoglobulin (Santa Cruz Biotechnology, Santa Cruz, CA, USA) diluted 1:10,000 in 1% (*w*/*v*) non-fat dried milk and TBS-T buffer at 25 °C for 1 h. After washing the membranes three times with TBS-T buffer for 10 min, the membranes were visualized using an Immobilon Western Chemiluminescent HRP Substrate (Millipore, Burlington, MA, USA), and signals were detected using a G: BOX Chemi XL (Syngene, Frederick, MD, USA).

## 4. Conclusions

In summary, our study highlights the significance of spermidine in facilitating the binding of IVT RNA to mesoporous silica. Both electrostatic interaction and hydrogen bonding between secondary amine groups of spermidine and silica surface are involved in the binding of RNA to mesoporous silica. Desorption of the silica-bound RNA in the chromatography setup was accomplished by elution with solutions containing salt, EDTA, or urea, among which EDTA was the most efficient reagent to recover the silica-adsorbed RNA. The spermidine-mediated adsorption of RNA to mesoporous silica and subsequent elution with EDTA was applicable for RNA purification from the post-in vitro transcription reaction solution without the need for a prior salt precipitation step. Importantly, a higher RNA recovery rate was observed with the mesoporous silica-based RNA purification with EDTA elution than with the ethanol elution of RNA in the cellulose-based spin column chromatography. Moreover, the RNA purification with EDTA elution in the spermidine-based mesoporous silica chromatography resulted in the efficient removal of at least 80% of dsRNA contaminants from the input IVT RNA. It would be valuable to conduct further research on optimizing the specificity of the method for different RNA sequences or structures, enhancing sample recovery, and minimizing potential interference from contaminants to ensure high-quality IVT RNA purification. Taken together, the mesoporous silica-based RNA adsorption/desorption provides a facile IVT RNA purification method with a higher recovery rate as well as a successful reduction of dsRNA content. Thus, our RNA purification chromatography system comprising mesoporous silica as an RNA adsorbent, spermidine as the linkage between RNA and silica, and EDTA as an elution reagent provides a promising approach for purification of intact IVT RNA, which can be used for various downstream applications in the mRNA production process.

## Figures and Tables

**Figure 1 ijms-24-12408-f001:**
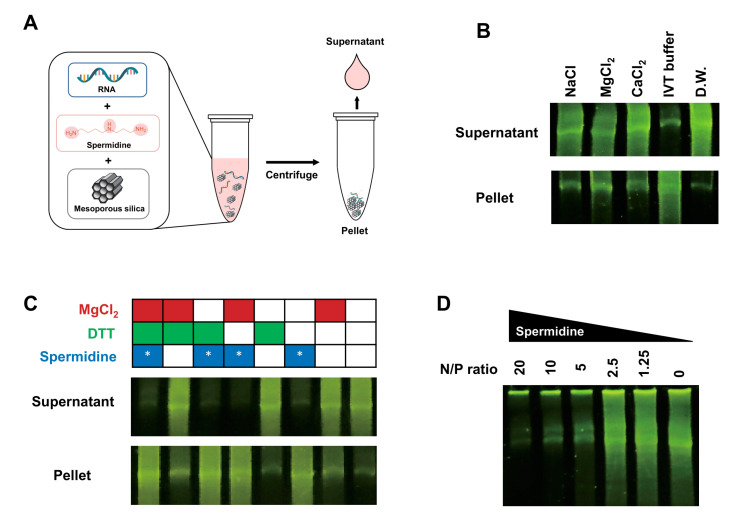
Spermidine in the in vitro transcription buffer promotes RNA binding to mesoporous silica. The LiCl-precipitated IVT RNA (2 μg) mixed with mesoporous silica particles was fractionated as supernatant and pellet by centrifugation. (**A**) Schematic illustration of the experimental procedure for determining adsorption of RNA onto mesoporous silica. (**B**) Adsorption of RNA to mesoporous silica in the presence of NaCl, MgCl2, CaCl2, distilled water (DW), and 1 × IVT buffer. Aliquots of the supernatant and pellet were analyzed by electrophoresis on denaturing PAGE (8 M urea, 6%), and RNA was stained with SYBR Gold and visualized by UV illumination. (**C**) Evaluation of RNA adsorption onto mesoporous silica in the presence of various combinations of the IVT buffer substances. Aliquots of the supernatant and pellet were analyzed for the presence of RNA, as performed in (**B**). Asterisk (*) indicates the inclusion of spermidine. (**D**) Examination of RNA adsorption onto mesoporous silica supplemented with different amounts of spermidine at various N/P ratios. Aliquots of the supernatant were analyzed by electrophoresis, as described in (**B**).

**Figure 2 ijms-24-12408-f002:**
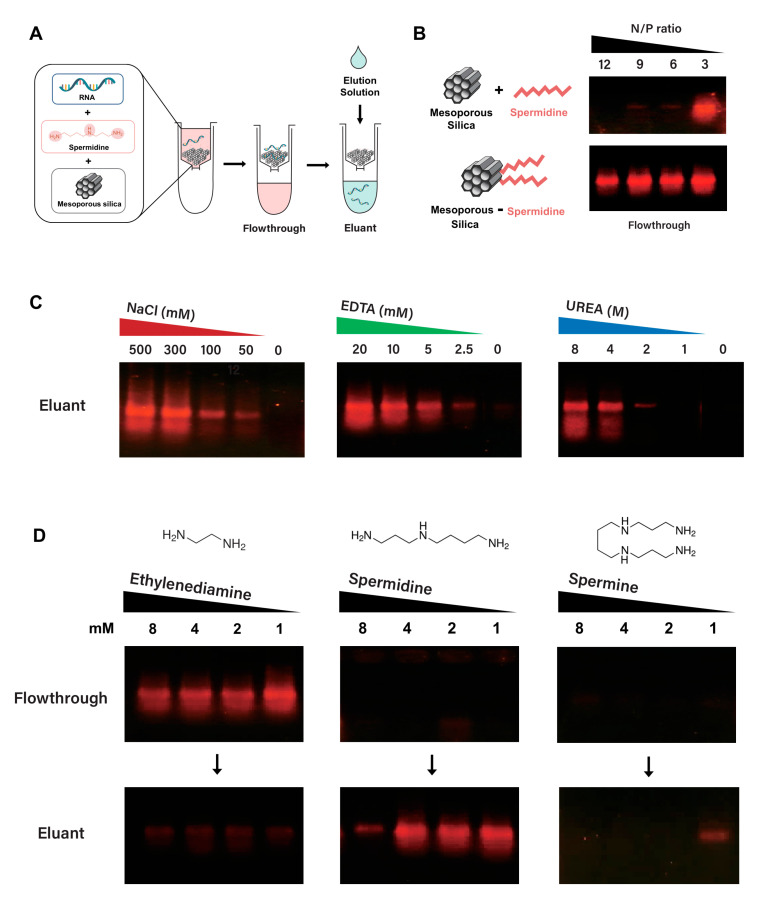
Assessment of mesoporous silica spin column chromatography; silica−spermidine interfaces, elution reagents, and polyamines for purification of IVT RNA. (**A**) Experimental process for determining the adsorption/desorption of IVT RNA in the spin column filled with mesoporous silica. (**B**) Comparison of RNA binding efficiency in the presence of spermidine at various N/P ratios; spermidine in the mobile phase vs. silica-conjugated spermidine in the stationary phase. The LiCl-precipitated IVT RNA (0.2 μg) mixed with or without spermidine at various N/P ratios was purified by a spin column filled with spermidine-conjugated mesoporous silica or mesoporous silica, respectively. Aliquots of flowthrough were analyzed by electrophoresis on a 1.0% agarose gel with GelRed-staining of RNA visualized by UV illumination. (**C**) Desorption of RNA from mesoporous silica with various eluting reagents; NaCl, EDTA, and urea. The LiCl-precipitated IVT RNA (2 μg) mixed with spermidine at an N/P ratio of 20 was purified by a spin column filled with mesoporous silica using an elution buffer containing various eluting reagents. Each eluant fraction was analyzed for the presence of RNA, as described in (**B**). (**D**) Binding and elution of RNA in the presence of various polyamines, such as ethylenediamine, spermidine, or spermine, at decreasing concentrations. The LiCl-precipitated IVT RNA (2 μg) mixed with each polyamine at varying concentrations was purified by a spin column filled with mesoporous silica using an elution buffer containing 300 mM NaCl. The presence of IVT RNA in each flowthrough and eluant fraction was analyzed as described in (**B**).

**Figure 3 ijms-24-12408-f003:**
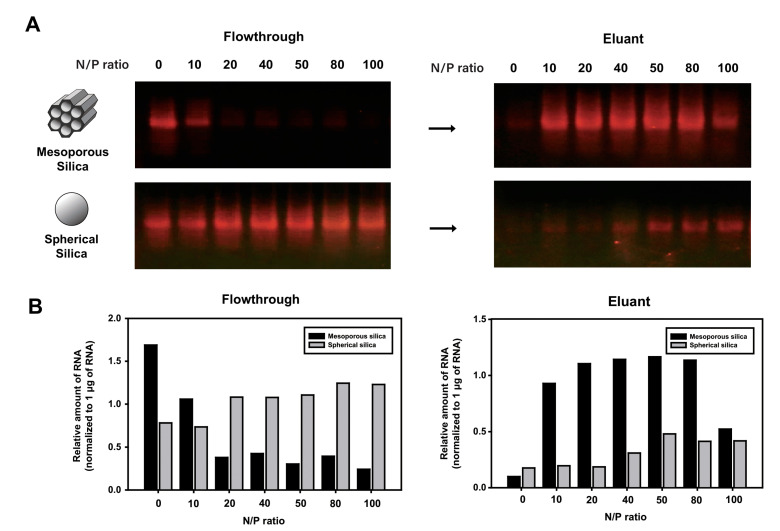
Evaluation of two types of silica particles as RNA adsorbent: mesoporous silica and spherical silica. The LiCl-precipitated IVT RNA (2 μg) mixed with spermidine at various N/P ratios was purified by a spin column filled with mesoporous silica or spherical silica using an elution buffer containing 300 mM NaCl. (**A**) Binding and elution of the IVT RNA using mesoporous silica or spherical silica in the presence of spermidine at an increasing N/P ratio. Aliquots of the fractionated RNA samples were analyzed on 1.0% agarose gel stained with GelRed and visualized by UV illumination. (**B**) The relative amounts of RNA present in each flowthrough (unbound RNA) and eluant (silica-bound RNA), which are shown in agarose gel (**A**), were quantified and normalized to 1 μg RNA loaded in the separate lane.

**Figure 4 ijms-24-12408-f004:**
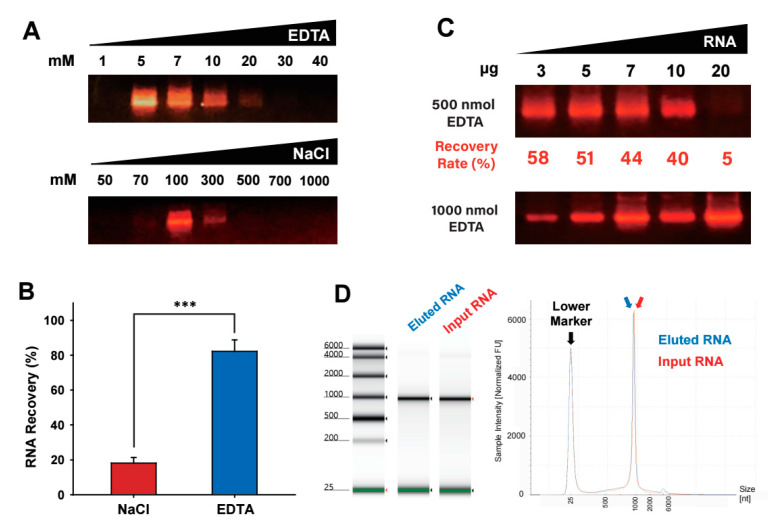
Evaluation of the elution reagents for recovery of silica-adsorbed RNA. The LiCl-precipitated IVT RNA (10 μg) was purified by spin column-based mesoporous silica chromatography using elution buffer containing either EDTA or NaCl with stepwise gradient concentrations. (**A**) Comparison of RNA recovery with salt (NaCl) and chelating reagent (EDTA) as an eluting reagent at a stepwise increase of concentration. Each eluant fraction was electrophoresed in 1.0% agarose gel pre-stained with GelRed, and RNA was visualized under UV illumination. (**B**) RNA recovery rate in each elution reagent was combined and normalized to the initial amount of input RNA. Data are presented as the mean ± standard deviation, *n* = 3; *** *p* < 0.005, NaCl vs. EDTA. (**C**) Various amounts of RNA were mixed with spermidine at a fixed N/P ratio of 20 and purified by spin column-based mesoporous silica chromatography by elution with 500 nmol and 1000 nmol EDTA. Each eluant fraction was electrophoresed in 1.0% agarose gel prestained with GelRed, and the RNA present in each fraction was visualized with UV transillumination. The recovery rates of eluants with 500 nmol EDTA were calculated by quantifying the amount of RNA using the RiboGreen assay. (**D**) Assessment of RNA quality in size, recovery rate, and intactness. The recovered RNA from the mesoporous silica-based chromatography through EDTA elution, as performed in (**A**), was analyzed by a capillary electrophoresis system, TapeStation. Gel-like electropherogram and elution profile were displayed for comparison of the input RNA and the EDTA-eluted RNA.

**Figure 5 ijms-24-12408-f005:**
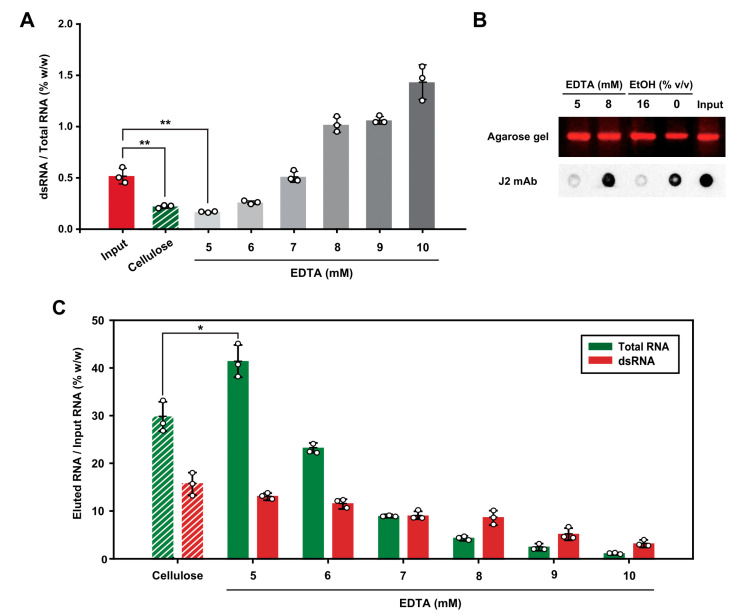
Removal of dsRNA from IVT RNA by spin column-based mesoporous silica chromatography. The unpurified IVT RNA (10 μg) was purified by spin column-based mesoporous silica chromatography using elution buffer containing EDTA with stepwise increasing concentration (5 to 10 mM). For comparison, the unpurified IVT RNA (10 μg), incubated with cellulose in the presence of chromatography buffer containing 16% (*v*/*v*) ethanol, was purified using microcentrifuge spin columns. (**A**) dsRNA content in each eluant was determined using the ELISA for dsRNA detection, and the dsRNA content was divided by the amount of total RNA in each eluant fraction. Bar graphs showing the relative amount of dsRNA in the total RNA are presented as the mean ± standard deviation, *n* = 3 (** *p* < 0.01). (**B**) Dot blotting analysis of dsRNA eluted either from the mesoporous silica with EDTA or the cellulose with ethanol. Images of dots blotted onto the paper were quantified for the relative presence of dsRNA in the sample. The corresponding amount of total RNA in each eluant was loaded on 1.0% agarose gel pre-stained with GelRed, and the RNA present in each fraction was visualized and quantified. (**C**) Proportions of total RNA and dsRNA in each eluant were analyzed. The amount of total RNA and dsRNA in the eluant was determined using the RiboGreen assay and the dsRNA ELISA, respectively. All experiments were repeated three times, and error bars display the standard deviation, *n* = 3 (* *p* < 0.05).

## Data Availability

Data that support the findings of this study are available within the article and Appendix A.

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
