# Peer review of "Mesoporous Silica Particle as an RNA Adsorbent for Facile Purification of In Vitro-Transcribed RNA"

_ijms, 2023, doi:10.3390/ijms241512408_

Round 1

Reviewer 1 Report

The paper entitled “Mesoporous silica particle as RNA adsorbent for facile purification of in vitro-transcribed RNA” contains a series of valuable results.

The paper is constructed well, the introduction contains enough information to recognize the usefulness of the topic and the reasons for doing the research. The binding of IVT RNA to mesoporous silica was studied and confirmed in detail. The research conducted to find the component that is responsible for the binding was planned correctly. The effect of compounds used was also tested, and the conditions to increase the binding affinity of spermidine were planned and confirmed correctly. The desorption studies were also done in detail.

The presence of secondary amine groups (internal as the authors mentioned) probably is the main reason for the found selectivity for spermine and spermidine compared to ethylenediamine, the authors” conclusion is correct. It would be worth studying the explanation why the secondary amine groups of spermine or spermidine have better binding ability compared to ethylenediamine  (line 182), e.g., with deuteration experiments (IR and solid phase NMR before and after deuteration). The strength of hydrogen bonds is not the same with H and D atoms towards surface OHs of silica. I suggest defining these amine groups as “secondary amine” groups and not “internal”, because their properties and not (only) their positions are responsible for their effect.  

Removal of dsRNA is a useful result, the experiments to clarify the conditions were carried out correctly. Overall, the paper is a good one, well planned, the research is built logically, the text is written well enough, and the results are good and useful.

I can suggest accepting it to publish In IJMS.

Author Response

We appreciate your thoughtful suggestion regarding our work on the selectivity of spermine and spermidine compared with ethylenediamine and the use of the term "internal amine" groups. We agree with your observation that the presence of secondary amine groups in spermine and spermidine likely plays a significant role in the observed selectivity. Thus, it is worthwhile to investigate the reasons for their better binding ability compared to ethylenediamine in a near future as a separate study.

Regarding the nomenclature, we acknowledge your suggestion that referring to these amine groups as "internal" might not adequately capture their properties responsible for the observed effect. As you suggested, we removed the term "internal" and replaced it with "secondary amine" into our revised manuscript (line 182, 195, 512), as it more accurately reflects their chemical nature and properties influencing the selectivity.

Once again, we thank you for your insightful comment, which has undoubtedly improved the clarity and understanding of our work.

Reviewer 2 Report

The researchers developed a method for purifying in vitro transcribed (IVT) RNA using mesoporous silica particles and spermidine. The adsorbed RNA could be easily desorbed using elution buffers containing salt, EDTA, or urea. The researchers demonstrated that different concentrations of EDTA in the elution buffer could remove at least 80% of double-stranded RNA (dsRNA) contaminants from the adsorbed RNA. However, there are several minor revisions that should be considered:

  1. In section 2.3, it would be helpful to include more details about the characterization of the mesoporous silica particles used in the study. This could include information about particle size, surface area, pore size, and any functionalization of the particles.

  2. The conclusion lacks a discussion on potential limitations or challenges associated with the proposed method. It would be valuable to address factors such as the potential impact of different RNA sequences or structures on dsRNA removal efficiency, the potential for sample loss during the purification process, or any potential interference from contaminants.

The quality of the English language in the provided information is generally good. However, there are a few instances where clarity and coherence could be improved. Minor revisions and proofreading would enhance the clarity and readability of the text.

Author Response

  1. In section 2.3, it would be helpful to include more details about the characterization of the mesoporous silica particles used in the study. This could include information about particle size, surface area, pore size, and any functionalization of the particles.

Thank you for your valuable comment regarding the characterization of the mesoporous silica particles used in our study. In consideration of your thoughtful suggestion, we have included a description of the mesoporous silica particles in section; ‘2.3. Mesoporous silica rather than spherical silica with EDTA elution as facile adsorption/desorption of RNA’ (line 202-205):

‘We employed mesoporous silica of the SBA-15 (Santa Barbara Amorphous-15) type, which is characterized by particles with a size of less than 150 μm and well-defined pores with a diameter of 12 nm. The highly ordered hexagonal pattern of these pores results in a surface area up to 1000 m²/g. [1]’

  1. The conclusion lacks a discussion on potential limitations or challenges associated with the proposed method. It would be valuable to address factors such as the potential impact of different RNA sequences or structures on dsRNA removal efficiency, the potential for sample loss during the purification process, or any potential interference from contaminants.

We are grateful for reviewer’s keen comments, which highlighted the potential limitations and challenges associated with our proposed method. Indeed, addressing these factors is necessary for further enhancing the applicability and reliability of our approach. Considering your valuable feedback, we have incorporated discussions on these aspects in the Conclusion section of our revised manuscript (line 523-526):

‘It would be valuable to conduct further research on optimizing the method's specificity for different RNA sequences or structures, enhancing sample recovery, and minimizing potential interference from contaminants to ensure high-quality IVT RNA purification.’

Comments on the Quality of English Language

The quality of the English language in the provided information is generally good. However, there are a few instances where clarity and coherence could be improved. Minor revisions and proofreading would enhance the clarity and readability of the text.

Thank you for your feedback on the quality of the English in the provided information. We have carefully revised the text, making the necessary revisions to enhance the clarity and coherence of the manuscript. Your guidance has undoubtedly contributed to improving the overall quality of our study and we are sincerely grateful for your support.

Once again, we thank you for your valuable suggestions, which have contributed significantly to the refinement of our manuscript and have enriched the quality of our study.

References

  1. Verma, P.; Kuwahara, Y.; Mori, K.; Raja, R.; Yamashita, H. Functionalized mesoporous SBA-15 silica: recent trends and catalytic applications. Nanoscale 2020, 12, 11333-11363, doi:10.1039/d0nr00732c.